# Low-frequency vibrational density of states of ordinary and ultra-stable glasses

Ding Xu [1,2,4], Shiyun Zhang [1,2,4], Hua Tong [2], Lijin Wang [3] ✉ & Ning Xu [1,2] ✉

A remarkable feature of disordered solids distinct from crystals is the violation of the Debye scaling law of the low-frequency vibrational density of states. Because the low-frequency vibration is responsible for many properties of solids, it is crucial to elucidate it for disordered solids. Numerous recent studies have suggested power-law scalings of the low-frequency vibrational density of states, but the scaling exponent is currently under intensive debate. Here, by classifying disordered solids into stable and unstable ones, we find two distinct and robust scaling exponents for non-phononic modes at low frequencies. Using the competition of these two scalings, we clarify the variation of the scaling exponent and hence reconcile the debate. Via the study of both ordinary and ultra-stable glasses, our work reveals a comprehensive picture of the low-frequency vibration of disordered solids and sheds light on the low-frequency vibrational features of ultra-stable glasses on approaching the ideal glass.

Low-temperature properties of solids, such as specific heat and thermal conductivity, are closely related to the excitation of low-frequency vibrational states. For crystals, it is well-established that the vibrational states, i.e., phonons, form a low-frequency vibrational density of states (VDOS) following the Debye scaling law: $D(\omega) \sim \omega^{d-1}$, where $\omega$ is the frequency and $d$ is the spatial dimension, resulting in the $T^d$ scaling of the specific heat at low temperatures $T$[1]. The thermal conductivity is believed to be governed by the specific heat, phonon mean free path, and sound velocity. In crystals, because the phonon mean free path and sound velocity remain approximately constant in temperature, the thermal conductivity follows the low-temperature scaling of the specific heat[1].

However, we face great challenges when dealing with disordered solids such as glasses. The low-temperature scalings of the specific heat and thermal conductivity are no longer $T^{d}$[2-4]. When $T < 1K$, the specific heat is linearly scaled with $T$[2-4], which is attributed largely to the existence of two-level systems instead of the VDOS[5,6]. It is also believed that the two-level systems change the mean free path, causing anomalous behaviors of the thermal conductivity. At higher temperatures, the VDOS matters. The disordered structure of glasses causes the coexistence of phonon-like and non-phononic modes at low frequencies[7-14], so the VDOS is at least a superposition of the Debye scaling and that of the non-phononic modes. The excess non-phononic modes form a peak in $D(\omega)/\omega^{d-1}$, defined as the boson peak[11,12,15]. It has been shown that the boson peak may be correlated with the simultaneity of the peak in $c_p/T^3$, with $c_p$ being the constant-pressure specific heat, and the plateau in the thermal conductivity at the boson peak temperature (~10$K$ for typical glasses such as vitreous silica)[4]. Both simulations and experimental measurements such as the neutron scattering and X-ray, have significantly advanced our understanding of the constituent modes of the boson peak[11-16]. However, what the VDOS of non-phononic modes looks like below the boson peak frequency is still an unsettled issue[7,8,17-31], which is crucial to understanding the thermal properties in the 1–10$K$ temperature regime. In addition to the thermal properties, the anomalous low-frequency non-phononic modes have been successfully applied to understand various other properties of disordered solids, e.g., mechanical failure[32-37], glass transition[13,38,39], and heterogeneous dynamics of glass-forming liquids[40-42].

[1]Hefei National Research Center for Physical Sciences at the Microscale and CAS Key Laboratory of Microscale Magnetic Resonance, University of Science and Technology of China, Hefei 230026, P. R. China. [2]Department of Physics, University of Science and Technology of China, Hefei 230026, P. R. China. [3]School of Physics and Optoelectronic Engineering, Information Materials and Intelligent Sensing Laboratory of Anhui Province, Anhui University, Hefei 230601, P. R. China. [4]These authors contributed equally: Ding Xu and Shiyun Zhang. ✉e-mail: lijin.wang@ahu.edu.cn; ningxu@ustc.edu.cn

Numerous recent studies suggest that the low-frequency VDOS of non-phononic modes exhibits the $\omega^\alpha$ scaling with $\alpha \neq d - 1$[7,8,17–31]. However, the value of the exponent $\alpha$ is still under debate. A popular argument is that $\alpha = 4$ for generic glasses[7,8,17–19], i.e., zero-temperature disordered solids, which are constrained well above isostaticity and are thus not governed by the jamming physics[43,44]. It has been claimed that the quartic scaling is independent of spatial dimensions[18,21] and interaction potentials[19] and is valid for low-temperature glasses as well[45]. There are theories supporting this scaling, e.g., mean-field theories based on replica[46,47] and effective medium approximation[48,49], and phenomenological theories[50–53]. However, some other studies also reported deviations of $\alpha$ from 4. It has been shown that $\alpha$ may vary with the glass stability[22,23], system size[20,21,24], stress distribution[25], and frequency range accessed[26–29]. There are also models arguing that $\alpha \neq 4$. For example, the fluctuating elasticity theory predicts $\alpha = d + 1$[54,55]; the fold instability argument predicts $\alpha \approx 3$[36], independent of spatial dimensions.

Note that generic glasses lie at local minima of the complex energy landscape[56], whose stabilities can vary a lot from each other. One can tell that the variation of $\alpha$ mentioned above is more or less related to the stability. However, the local minima with various degrees of stability were always mixed up to calculate the VDOS in previous studies. Moreover, probably limited by the development of experimental techniques, as far as we know, there have been no direct experimental measurements of $\alpha$ for molecular glasses. Therefore, the examination of $\alpha$ has heavily relied on simulations. In most of previous simulations, the VDOS was calculated for systems with periodic boundary conditions, whose shapes were not allowed to change. However, it has been shown that some glasses that are stable under periodic boundary conditions may be unstable under certain deformations[57,58]. Apparently, the effects of such deformation stability on the VDOS were completely overlooked.

Here, we systematically study the low-frequency VDOS for both ordinary and ultra-stable model glasses quenched from different parent temperatures $T_p$. Remarkably different from previous approaches, we divide all glasses into two categories: stable ones, which can resist any infinitesimal deformations, and unstable ones, which are unstable subject to some infinitesimal deformations, and calculate their VDOSs separately. The VDOSs for stable and unstable solids depart from each other below a crossover frequency $\omega_d$, where they have different scaling exponents. For unstable solids, $\alpha = \alpha_u \approx 3.3$, independent of system size and spatial dimension. For stable solids, $\alpha = \alpha_s \approx 5.5$ and 6.5 in 2D ($d = 2$) and 3D ($d = 3$), respectively, which does not vary with system size either. The superposition of these two VDOSs results in the VDOS studied in previous approaches. This explains the variation of $\alpha$ under various circumstances. Moreover, we observe the emergence of an $\omega^4$ scaling right above the $\omega^{\alpha_s}$ one when the system size of stable solids increases for both ordinary and ultra-stable glasses in 3D. Interestingly, our results suggest that the number of non-phononic modes forming the $\omega^{\alpha_s}$ and $\omega^4$ scalings decays with the decrease of $T_p$, possibly vanishing at a sufficiently low $T_p$. Therefore, our study may shed light on the perspective of the vibrational features of the ideal glass.

## Results

In this work, we mainly show results for systems composed of polydisperse soft particles interacting via the inverse-power-law (IPL) potential (see Methods for details), which have been widely used to study the glass transition[8,26,28,59–62]. In Supplementary Fig. 1 of the Supplementary Information and a parallel study, we also show consistent results for Lennard–Jones and harmonic potentials, suggesting the generality of our findings. We obtain the zero-temperature glasses by instantaneously quenching liquids equilibrated at the parent temperature $T_p$. It is well-known that the stability of quenched glasses increases with the decrease of $T_p$ when $T_p$ is lower than the onset

temperature $T_{on}$, i.e., the crossover temperature from Arrhenius to super-Arrhenius dynamics[56]. We will first study glasses obtained from a given $T_p$ and discuss the $T_p$ dependence afterward.

## VDOSs for stable and unstable solids

In most of the previous simulations, the normal modes of vibration were obtained from the diagonalization of the normal Hessian matrix, with the elements being the second derivatives of the potential energy with respect to particle coordinates. No boundary deformation was taken into account in such an approach. A glass was treated as a stable one if all nontrivial eigenvalues of the normal Hessian matrix were positive. However, this cannot guarantee that the glass is stable subject to boundary deformations. If we introduce the $d(d+1)/2$ degrees of freedom corresponding to the boundary deformations (shear and compression) and construct the extended Hessian matrix (see Methods), the matrix of some glasses may have negative eigenvalues, indicating that the glasses are unstable under some deformations. We thus define these glasses as unstable glasses. On the other hand, the glasses whose extended Hessian matrix has no negative eigenvalues are defined as stable glasses. Note that the extended Hessian matrix is only used to classify all glasses into stable and unstable ones, and VDOS is still calculated from the normal Hessian matrix. Here, we denote $D_s(\omega)$, $D_u(\omega)$, and $D(\omega)$ as the VDOSs of stable, unstable, and all glasses, respectively.

Figure 1a, b compares $D(\omega)$, $D_s(\omega)$, and $D_u(\omega)$ in 2D and 3D, respectively. They collapse above a crossover frequency $\omega_d$, and depart from each other otherwise. Both low-frequency tails of $D_s(\omega)$ and $D_u(\omega)$ display a clear power-law scaling behavior, $D_s(\omega) \sim \omega^{\alpha_s}$ and $D_u(\omega) \sim \omega^{\alpha_u}$. Beyond that, $D_u(\omega)$ forms a valley bottomed at $\omega_d$, while $D_s(\omega)$ still monotonically increases and transits to $\omega_d$. However, $\alpha_s$ and $\alpha_u$ are apparently different. In 2D and 3D, $\alpha_s = 5.5 \pm 0.2$ and $6.5 \pm 0.2$, respectively. In contrast, $\alpha_u = 3.3 \pm 0.1$ in both 2D and 3D. This $\alpha_u$ value is close to the $\alpha \approx 3$ arguments of the fold instability model[36]. Note that $\alpha \approx 3$ is obtained based on the approximation that the distribution of the stress distance to instabilities is constant[36], which may fluctuate if the distribution is not strictly flat. The fold instability model is raised

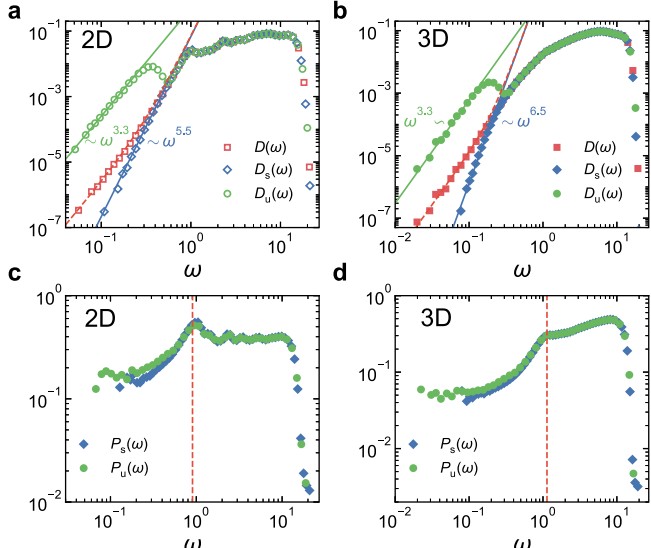

**Fig. 1 | Comparison of VDOS and participation ratio of stable, unstable, and all glasses. a** VDOSs of 2D systems with $N = 256$ and $T_p = 0.12$. **b** VDOSs of 3D systems with $N = 1000$ and $T_p = 0.18$. The solid lines are power-law fittings to $D_s(\omega)$ and $D_u(\omega)$ at low frequencies. The red dashed lines are results from Eq. (1). They are in excellent agreement with the simulated $D(\omega)$ at low frequencies. **c** and **d** show the participation ratio of stable and unstable solids for the same systems in (**a**) and (**b**), respectively. The vertical dashed lines mark the frequency of the first Goldstone mode.

for glasses with weak stability and is prone to rearrangement upon deformations and does not rely on spatial dimension. This agreement thus proposes a plausible physical origin of the scaling behavior of $D_u(\omega)$.

By definition, the low-frequency part of $D(\omega)$ should be the superposition of $D_s(\omega)$ and $D_u(\omega)$:

$$
\begin{aligned}
D(\omega) &= f_s D_s(\omega) + (1 - f_s) D_u(\omega) \\
&= f_s A_s \omega^{\alpha_s} + (1 - f_s) A_u \omega^{\alpha_u},
\end{aligned}
\tag{1}
$$

where $f_s$ is the fraction of stable glasses, and $A_s$ and $A_u$ are prefactors of $D_s(\omega)$ and $D_u(\omega)$, respectively. In Fig. 1a, b, we compare the simulated $D(\omega)$ with the prediction by Eq. (1) (dashed line). They are in excellent agreement at low frequencies.

As done in previous studies, the low-frequency part of $D(\omega)$ can be fitted with $\omega^\alpha$. Figure 1a, b indicates that $\alpha$ should be between $\alpha_u$ and $\alpha_s$, if we perform the fitting. Now, the excellent agreement between $D(\omega)$ and Eq. (1) provides another interpretation of the $\alpha$ value at the low-frequency tail. If the values of $\alpha_s$ and $\alpha_u$ are definite, the $\alpha$ value is jointly determined by $f_s$, $A_s$, and $A_u$, which may change with parameters such as system size and parent temperature. We are thus able to understand why $\alpha$ was reported to vary under some circumstances[20-24]. Moreover, at low enough frequencies, $D_u(\omega)$ dominates. This may be the reason why lower values of $\alpha$ were always observed when rather low-frequency regimes were accessed[24,27,28].

Figure 1c, d compares the participation ratio, $P_s(\omega)$ and $P_u(\omega)$, of stable and unstable solids. A mode with a lower participation ratio is more localized. We can see that, below the first Goldstone (phonon-like) mode, the low-frequency modes forming the $\omega^{\alpha_s}$ and $\omega^{\alpha_u}$ scalings have the lowest participation ratios and are thus most quasi-localized on average. However, the degrees of quasi-localization of stable and unstable solids are similar, only that unstable solids extend to lower frequencies.

Figure 2a, b visualizes the structures of the modes lying in the $\omega^{\alpha_u}$ and $\omega^{\alpha_s}$ scaling regimes. They both exhibit the typical feature of quasi-localized modes with localized regions hybridizing with the plane-wave-like background. For the unstable solid in Fig. 2a, we show in Fig. 2c its unstable mode of the extended Hessian matrix, whose eigenvalue is negative. It looks almost identical to the mode in Fig. 2a. The dot product of the two normalized modes in Fig. 2a, c is 0.997. Note that when a disordered solid approaches the fold instability under load such as shear and compression, its lowest-frequency mode is responsible for the instability, whose frequency decays to zero following a power law while its structure remains unchanged[36]. This type of mode contributes to the $\omega^3$ behavior predicted by the fold instability argument[36]. Therefore, the perfect agreement between the lowest-frequency mode of the unstable solid and the unstable mode of the extended Hessian matrix is the evidence supporting our argument that $\alpha_u \approx 3.3$ originates from fold instabilities. Figure 2d illustrates how the boundary deforms associated with the unstable mode of the extended Hessian matrix shown in Fig. 2c. It involves both shear and

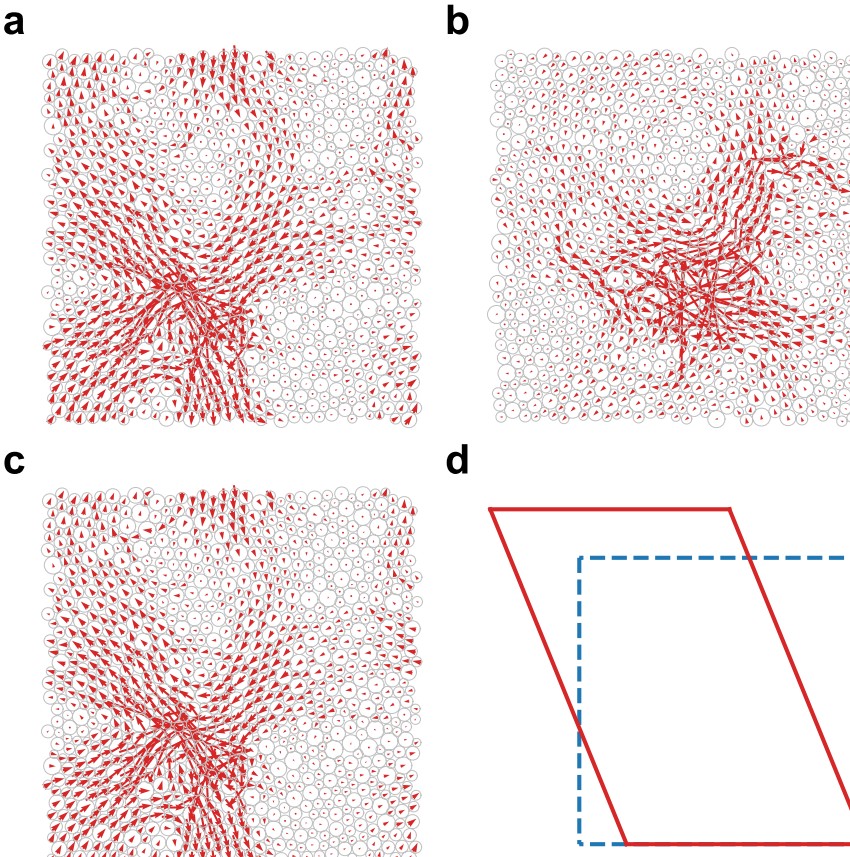

**Fig. 2 | Visualization of the lowest-frequency modes of stable and unstable solids. a** Structure of the lowest-frequency mode of an unstable solid. **b** Structure of the lowest-frequency mode of a stable solid. Here, we show 2D examples with $N = 1024$ and $T_p = 0.25$. The red arrows show the polarization vectors of particles. The modes lie in the $\omega^{\alpha_u}$ and $\omega^{\alpha_s}$ regimes, respectively. **c** Structure of the unstable mode with the lowest and negative eigenvalue of the extended Hessian matrix for the same unstable solid in (**a**). It looks almost identical to that in (**a**). The dot production of the normalized polarization vectors in (**a**) and (**c**) is 0.997. **d** Illustration of the boundary deformation associated with the unstable mode in (**c**). The ratio of three strains (see Methods) is $\epsilon_{xx} : \epsilon_{yy} : \epsilon_{xy} = -0.359 : 0.352 : -1$. The deformation involves both shear and compression (expansion).

compression, which is the typical form of the boundary deformation of unstable modes.

## System size dependence

Recently, it was reported that the value of $\alpha$ for $D(\omega)$ increased with the growth of system size for ordinary glasses quenched from high parent temperatures[20,21,24]. As shown in Fig. 3a, for 2D systems, $\alpha$ indeed grows from 3.4 to 4 when the system size $N$ changes from 256 to 4096. Interestingly, Fig. 3b, c shows that $\alpha_s$ and $\alpha_u$ remain constant in $N$. However, both $A_s$ and $A_u$ grow with $N$. Meanwhile, $f_s$ increases when $N$ increases, which can be fitted well with $1 - f_s \sim N^{-1.4}$, as illustrated in Fig. 3d. Therefore, the system size dependence of $\alpha$ in 2D directly reflects the competition among $f_s$, $A_s$, and $A_u$.

Figure 3 e–h indicates that similar system size evolution happens in 3D. When system size increases, $\alpha$ gradually increases. Again, $\alpha_s$ and $\alpha_u$ are insensitive to the change in system size, while $A_s$, $A_u$, and $f_s$ grow with $N$. However, the comparison between Fig. 3b, f demonstrates a seeming difference between 2D and 3D. In 2D, the $\omega^{\alpha_s}$ scaling extends all the way to the crossover frequency $\omega_d$, above which $D_s(\omega)$ and $D_u(\omega)$ collapse. In 3D, the $\omega^{\alpha_s}$ scaling is deviated above another crossover frequency $\omega_s < \omega_d$. Figure 3f shows that, when system size increases, $\omega_s$ decreases so that the frequency regime for the $\omega^{\alpha_s}$ scaling to survive is suppressed.

In addition to $\omega_d$ and $\omega_s$, there is another characteristic frequency $\omega_p < \omega_d$ of the first peak in $D_u(\omega)$. In Fig. 3d, h, we show the system size dependence of these three characteristic frequencies. In both 2D and 3D, $\omega_d$ is approximately scaled with $N^{-1/d}$. Since $D_s(\omega)$ and $D_u(\omega)$ deviate below $\omega_d$, if such system size dependence persists on approaching the thermodynamic limit, we would expect that the $\omega^{\alpha_s}$ and $\omega^{\alpha_u}$ scalings tend to disappear so that $D_s(\omega)$ and $D_u(\omega)$ eventually become identical to $D(\omega)$. Note that the Goldstone modes have the same system size dependence. It may be plausible to ask whether $\omega_d$ is associated with some inherent properties of disordered solids such as the elastic moduli, which contribute to the Goldstone modes. Figure 3d, h shows that $\omega_p$ is approximately scaled with $N^{-0.55}$ in both 2D and 3D. As seen from Fig. 3h, $\omega_s(N)$ in 3D roughly agrees with $\omega_p(N)$. At the current stage, we are not able to confirm whether there are any physical origins

of these characteristic frequencies and hope to leave them to future investigations.

In Fig. 4, we collapse the low-frequency parts of $D_s(\omega)$ and $D_u(\omega)$ for different system sizes by plotting $N^{-\nu_s}D_s(\omega)$ and $N^{-\nu_u}D_u(\omega)$ against $\omega N^{\nu_s}$ and $\omega N^{\nu_u}$, respectively. These scalings conserve the integrals of the VDOSs. Our best data collapse gives $\nu_s \approx 0.21$ for both 2D and 3D and $\nu_u \approx 0.35$ and $0.28$ for 2D and 3D, respectively. The scaling collapse indicates that $A_s \sim N^{(\alpha_s+1)\nu_s}$ and $A_u \sim N^{(\alpha_u+1)\nu_u}$, respectively.

Seen from Fig. 3f, the $\omega > \omega_s$ part of $D_s(\omega)$ in 3D shows the trend to converge to a master curve when system size increases. Figure 3f also indicates that $D_s(\omega)$ reaches the maximum at $\omega = \omega^* \approx 5$, above which $D_s(\omega)$ is plateau-like and gradually decreases. In Fig. 5a, we focus on $\omega < \omega^*$ with more system sizes. There seem to be three consecutive frequency regimes with different scalings: (i) $\omega^{\alpha_s}$ when $\omega < \omega_s$, (ii) $\omega^{\alpha_1}$ when $\omega_s < \omega < \omega_0$, and (iii) $\omega^{\alpha_2}$ when $\omega_0 < \omega < \omega^*$. Unlike the size-independent $\alpha_s$, $\alpha_1$ and $\alpha_2$ evolve with $N$. For the largest system sizes studied here, we can observe the emergence of $\alpha_1 \approx 4$ and $\alpha_2 \approx 1.5$. Right below the plateau of the VDOS ($\omega_0 < \omega < \omega^*$), mean-field theories predict an $\omega^2$ behavior due to marginal stability[63,64]. For the system sizes studied here, $\alpha_2$ slightly varies with system size. Although we are not able to exclude the possibility that $\alpha_2$ could approach 2 in sufficiently large systems, $\alpha_2 \approx 1.5$ observed here is still apparently lower than the mean-field value. It thus remains a question whether $\alpha_2$ is meaningful and related to marginal stability. Recent studies suggest that quasi-localized modes below $\omega_0$ could form the $\omega^4$ scaling. Here, we see this scaling right above $\omega_s$. However, whether this scaling is real or is just a crossover still needs to be examined in sufficiently large systems with good statistics. Note that, even if $\omega^4$ could be real, our results suggest that it does not generally exist. As shown in Fig. 3b, in 2D, there is no sign for the $\omega^4$ behavior to emerge in $D_s(\omega)$.

Assuming that the system size evolution of $\omega_s$ is still valid in much larger systems, we can expect that there is always a contribution of the $\omega^{\alpha_s}$ scaling below $\omega_s$, as long as the system size is finite. Therefore, the low-frequency tail of $D(\omega)$ is always jointly determined by $D_s(\omega)$ and $D_u(\omega)$ according to Eq. (1).

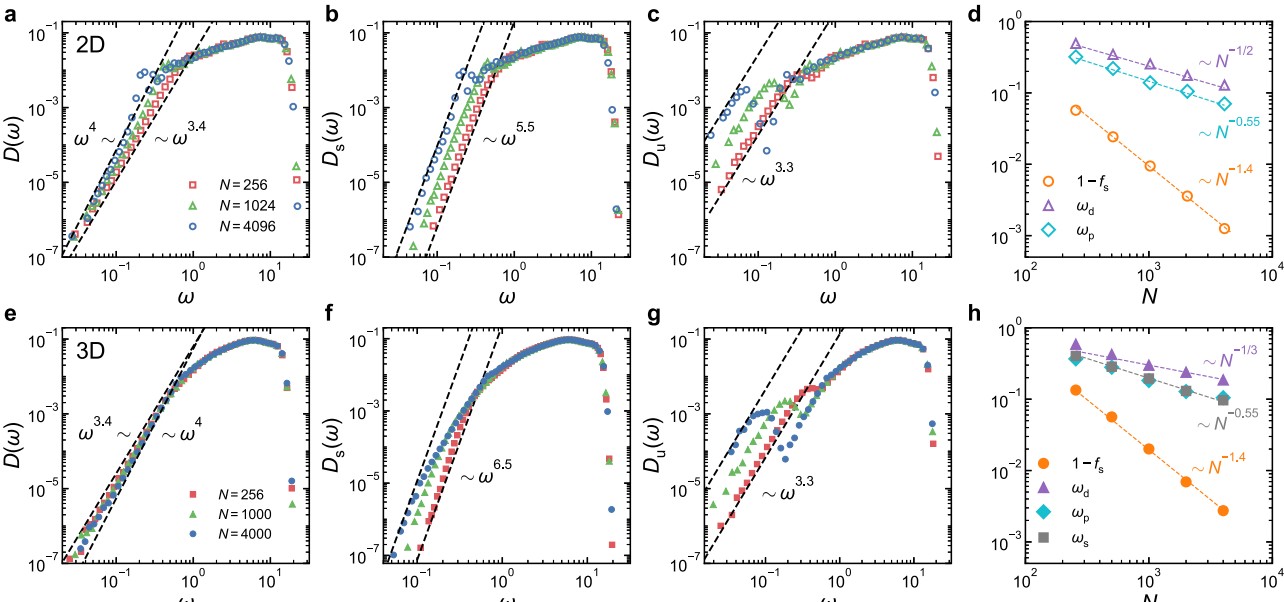

**Fig. 3 | System size dependence of the VDOSs. a–d** VDOSs of all stable and unstable glasses, $D(\omega)$, $D_s(\omega)$, and $D_u(\omega)$, in 2D and system size evolution of the fraction of stable glasses $f_s$, the frequency $\omega_p$ of the first peak in $D_u(\omega)$, and the frequency $\omega_d$ below which $D_u(\omega)$ and $D_s(\omega)$ depart from each other, respectively. **e–h** Results in 3D. In (**h**), we also show the system size depends of the frequency $\omega_s$ below which the $\omega^{\alpha_s}$ scaling exists. The parent temperature $T_p$ is approximately the onset temperature $T_{on}$ on both 2D and 3D systems. The dashed lines show the power-law scalings.

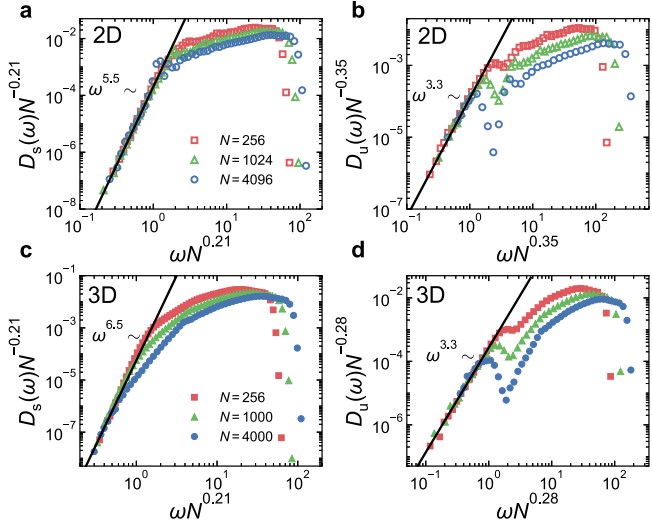

**Fig. 4 | Scaling collapse of the low-frequency parts of the VDOSs for different system sizes.** The VDOSs of stable and unstable glasses, $D_s(\omega)$ and $D_u(\omega)$, collapse at low frequencies, when $D_s(\omega)N^{-\nu_s}$ and $D_u(\omega)N^{-\nu_u}$ are plotted against $\omega N^{\nu_s}$ and $\omega N^{\nu_u}$, respectively. Results of 2D glasses are shown in (**a**) and (**b**), while (**c**) and (**d**) show results of 3D glasses. Here $\nu_s = 0.21$ for both 2D and 3D; $\nu_u = 0.35$ and $0.28$ for 2D and 3D, respectively. The solid lines are power-law fittings to the collapsed curves.

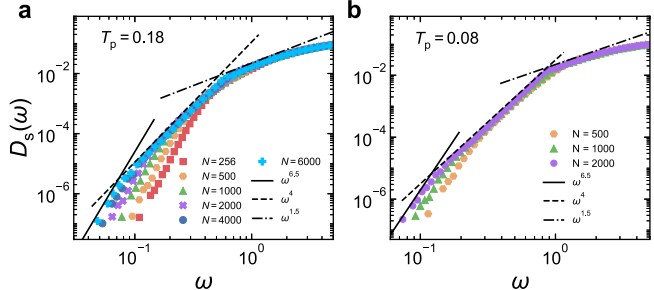

**Fig. 5 | System size evolution of the VDOS of stable glasses in 3D. a** $D_s(\omega)$ of ordinary glasses quenched from $T_p = 0.18$. **b** $D_s(\omega)$ of ultra-stable glasses quenched from $T_p = 0.08$. The solid, dashed, and dot-dashed lines show the $\omega^{\alpha_s}$, $\omega^4$, and $\omega^{1.5}$ scalings, respectively.

### Parent temperature dependence

It was also reported that $\alpha$ grew when the parent temperature $T_p$ decreased[22,23]. To understand this $T_p$ dependence, we study the VDOSs of glasses quenched from various $T_p$ ranging from above the onset temperature $T_{on}$ to near the glass transition temperature $T_g$. Figure 6 compares $D(\omega)$, $D_s(\omega)$, and $D_u(\omega)$ near the three representative temperatures, $T_{on}$, $T_{mc}$ (mode-coupling temperature), and $T_g$, for both 2D and 3D glasses. Figure 6a, e shows that $\alpha$ evolves roughly from 3.4 to 4 when $T_p$ decreases from $T_{on}$ to $T_g$ in both 2D and 3D, consistent with previous studies[22,24] and similar to the evolution with system size. The difference is that the low-frequency part of $D(\omega)$ apparently decays with the decrease of $T_p$. Figure 6b, c (f and g) indicates that $\alpha_s$ and $\alpha_u$ also remain constant in $T_p$ in 2D (3D). Unlike the system size dependence, $A_u$ is insensitive to the change of $T_p$. Therefore, the evolution of $D(\omega)$ at low frequencies is jointly determined by $A_s$ and $f_s$. When $T_p$ decreases, Fig. 6b, f shows that $A_s$ decreases, while $f_s$ increases, as shown in Fig. 6d, h.

For 3D ultra-stable glasses quenched from $T_p \approx T_g$, Fig. 5b shows that the three frequency regimes in $D_s(\omega)$ discussed above also emerge. Compared to the high $T_p$ case, the intermediate $\omega^4$ scaling seems to be more pronounced. For example, for smaller systems with $N \lesssim 1000$, there is no apparent $\omega^4$ scaling in Fig. 5a, but we can already see it in Fig. 5b. Again, the authenticity of the $\omega^4$ scaling needs to be verified in sufficiently large systems, which is however still absent in 2D ultra-stable glasses.

Figure 7 directly displays the $T_p$ dependence of $A_s$. For 3D systems, we also show the prefactor $A_4$ of the $\omega^4$ scaling above $\omega_s$. Both $A_s$ and $A_4$ keep decreasing with the decrease of $T_p$. Similar $T_p$ dependence was reported for the prefactor of the $\omega^4$ scaling of $D(\omega)$[8,31]. At the current stage, it is difficult to obtain reliable results at much lower $T_p$. If such $T_p$ dependence persists at even lower $T_p$, the number of low-frequency non-phononic modes significantly decreases and could be expected to vanish at low enough $T_p$. If this is the case, such low-temperature ultra-stable glasses will only have phonon-like modes at low frequencies. Although structurally disordered, evaluated by conventional criteria, the glasses could behave like crystals at long wavelengths. In fact, there was experimental evidence of the low-temperature Debye scaling for ultra-stable glasses[65–67], supporting that the non-phononic mode contribution can be negligible if the glass reaches the highest stability. It is thus interesting to figure out whether such ultra-stable glasses are prototypes of ideal glasses and under what temperatures they could exist.

## Discussion

By classifying disordered solids into stable and unstable ones, we find that their VDOSs, $D_s(\omega)$ and $D_u(\omega)$, depart from each other when $\omega < \omega_d$, with the low-frequency tails following distinct scaling laws, $D_s(\omega) \sim \omega^{\alpha_s}$ and $D_u(\omega) \sim \omega^{\alpha_u}$, respectively. The robustness of the values of $\alpha_s$ and $\alpha_u$ is verified by the solids with different sizes and quenched from different parent temperatures. Using this classification, we can understand the variation of the scaling exponent $\alpha$ reported previously. For finite-size disordered solids, it is due to the existence of unstable solids and the competition among the fraction of (un)stable solids and prefactors of the two scalings. Because unstable solids are inevitable in confined systems, our study can explain a recent experimental observation of $\alpha \approx 3$ in a confined quasi-2D nanosystem[68].

We also find that when system size increases, $\omega_d$ decreases so that the two scalings are pushed to lower frequencies. For stable solids in 3D, the $\omega^{\alpha_s}$ scaling only exists below another crossover frequency $\omega_s < \omega_d$, which also decreases with the increase of system size. When $\omega > \omega_s$, our results show the trend of the emergence of the $\omega^4$ scaling in the largest systems studied and ultra-stable glasses. At the current stage, we cannot confirm the authenticity of the $\omega^4$ scaling, which requires the verification of sufficiently large systems in future studies. For stable solids in 2D, we do not see the emergence of the $\omega^4$ scaling. Moreover, the prefactors of the $\omega^{\alpha_s}$ and $\omega^4$ scalings both decrease with the decrease of parent temperature, implying the possible existence of ultra-stable glasses with only crystal-like low-frequency vibrations at low enough temperatures. Such glasses may act as prototypes of the ideal glass.

In this work, we are focused on generic glasses, which are constrained well above isostaticity[43,44,69,70]. Marginally jammed solids near isostaticity are less stable than generic glasses concerned here. It is thus interesting to know whether and to what extent our findings here are applicable to marginally jammed solids. There are mean-field theories proposing the $\alpha = 2$ scaling of the VDOS[63,64]. The competition between different theoretical frameworks may complicate the vibrational features of marginally jammed solids moving away from the jamming transition[43,44]. We leave these discussions to a separate study.

## Methods
### Simulation model
Our systems contain $N$ polydisperse particles in a simulation cell with side length $L$ and periodic boundary conditions in all directions. All particles have the same mass $m$. Particles $i$ and $j$ interact via the IPL

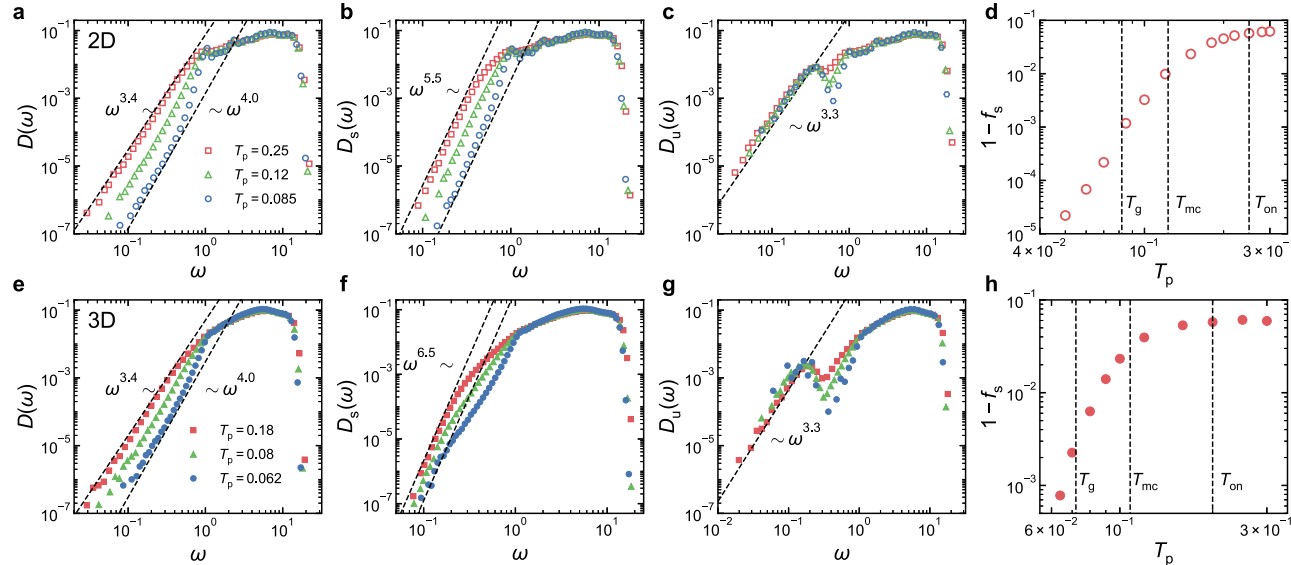

**Fig. 6 | Parent temperature dependence of the VDOSs. a–d** VDOSs of all stable, and unstable glasses, $D(\omega)$, $D_s(\omega)$, and $D_u(\omega)$, and parent temperature evolution of the fraction of stable glasses $f_s$ for $N = 256$ systems in 2D. **e–h** Results for

$N = 1000$ systems in 3D. The vertical dashed lines in (**d**) and (**h**) locate $T_{on}$, $T_{mc}$, and $T_g$, respectively. The dashed lines in the other panels show the power-law scalings.

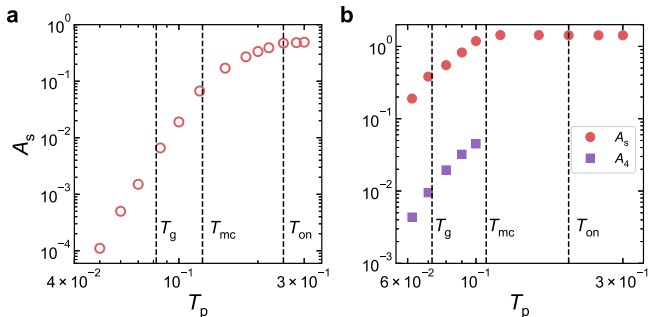

**Fig. 7 | Parent temperature dependence of prefactors of the VDOS of stable glasses. a** Prefactor $A_s$ versus $T_p$ in 2D. **b** Prefactors $A_s$ and $A_4$ versus $T_p$ in 3D. The vertical dashed lines locate $T_{on}$, $T_{mc}$, and $T_g$, respectively.

potential:

$$U(r_{ij}) = \left(\frac{\sigma_{ij}}{r_{ij}}\right)^{12} + c_0 + c_2 \left(\frac{r_{ij}}{\sigma_{ij}}\right)^2 + c_4 \left(\frac{r_{ij}}{\sigma_{ij}}\right)^4, \quad (2)$$

when their separation $r_{ij} \leq 1.25\sigma_{ij}$, and zero otherwise. The coefficients $c_0$, $c_2$, and $c_4$ ensure the continuity of the potential up to the second derivative at the cutoff. The particle diameter $\sigma$ is extracted from a continuous distribution $P(\sigma) = A\sigma^{-3}$, where $A$ is the normalization factor and $\sigma \in [\sigma_m, \sigma_M]$ with $\sigma_m/\sigma_M = 0.4492$. To enhance the glass-forming ability, we adopt a non-additive mixing rule to determine $\sigma_{ij}$ in Eq. (2):

$$\sigma_{ij} = \frac{\sigma_i + \sigma_j}{2}\left(1 - \epsilon|\sigma_i - \sigma_j|\right), \quad (3)$$

where $\epsilon$ measures the degree of non-additivity. We choose $\epsilon = 0.2$ to achieve a better performance[59].

We set the average particle diameter $\bar{\sigma}$, particle mass $m$, and the Boltzmann constant $k_B$ to be 1. The number density $\rho = N/L^d$ is 1.01 and 1.0 for 2D and 3D, respectively.

We use an efficient swap Monte Carlo algorithm[59] to prepare well-equilibrated liquids at parent temperatures $T_p$. The onset, mode-

coupling, and glass transition temperatures for our IPL model systems are $T_{on} \approx 0.25(0.20)$, $T_{mc} \approx 0.123(0.108)$, and $T_g \approx 0.082(0.072)$ in 2D (3D), respectively[8,60]. After equilibration at the parent temperature $T_p$, the liquids are rapidly quenched to zero temperature to obtain the zero-temperature glasses (inherent structures) via the fast inertial relaxation engine algorithm[71].

## Vibrational quantities

We consider two types of Hessian matrix. The normal Hessian matrix is defined as

$$M_n = \frac{\partial^2 U}{\partial \mathbf{R}^2}, \quad (4)$$

where $\mathbf{R} = (\mathbf{r}_1, \mathbf{r}_2, ..., \mathbf{r}_N)$ with $\mathbf{r}_i$ $(i = 1, 2, ..., N)$ being the location of particle $i$. The normal Hessian matrix does not take any boundary deformation into account. In comparison, the extended Hessian matrix with $(dN + n_{ex}) \times (dN + n_{ex})$ dimensions is[58]

$$M_e = \frac{\partial^2 U}{\partial \tilde{\mathbf{R}}^2}, \quad (5)$$

where $n_{ex} = d(d + 1)/2$ is the extra degrees of freedom of the system and $\tilde{\mathbf{R}} = (\mathbf{r}_1, \mathbf{r}_2, ..., \mathbf{r}_N, \epsilon_1, \epsilon_2, ..., \epsilon_{n_{ex}})$ with $\epsilon_i (i = 1, 2, ..., n_{ex})$ being the strain of the $i - \text{th}$ deformation. The strains $\epsilon_i$ are upper triangular elements of the $d \times d$ strain tensor

$$\begin{pmatrix} \epsilon_{\beta_1\beta_1} & \epsilon_{\beta_1\beta_2} & \cdots & \epsilon_{\beta_1\beta_d} \\ \epsilon_{\beta_2\beta_1} & \epsilon_{\beta_2\beta_2} & \cdots & \epsilon_{\beta_2\beta_d} \\ \vdots & \vdots & \ddots & \vdots \\ \epsilon_{\beta_d\beta_1} & \epsilon_{\beta_d\beta_2} & \cdots & \epsilon_{\beta_d\beta_d}, \end{pmatrix} \quad (6)$$

where $\beta_j$ $(j = 1, 2, ..., d)$ denotes the Cartesian coordinates. These $n_{ex}$ degrees of freedom involve boundary deformations, including compression (expansion) and shear. More details and the stability analysis using the extended Hessian matrix can be found in Ref. 58. The normal modes of vibration are obtained by diagonalizing the matrix using the Intel Math Kernel Library[72]. If the extended Hessian matrix has negative eigenvalues, the system is unstable to certain boundary

deformations[58]. Otherwise, the system is stable to compression and shear in arbitrary directions. The participation ratio of a normal mode $j$ is calculated as

$$P_j = \frac{\left(\sum_{i=1}^{N} \left|\mathbf{e}_i^j\right|^2\right)^2}{N \sum_{i=1}^{N} \left|\mathbf{e}_i^j\right|^4}, \tag{7}$$

where $\mathbf{e}_i^j$ is the polarization vector of particle $i$ in mode $j$. In the calculation of the VDOSs and the participation ratio, we exclude some lowest-frequency localized modes caused by rattler-like particles, as explained in Supplementary Fig. 2 of the Supplementary Information.

## Data availability
The data that support the findings of this study are included in the article and/or the Supporting Information and are available from the corresponding authors upon request.

## Code availability
The computer codes of this study are available from the corresponding authors upon request.

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

## Acknowledgements

We thank A.J. Liu for useful discussions. D.X., S.Z., H.T., and N.X. acknowledge the support from the National Natural Science Foundation of China (Grant Nos. 12334009, 12074355, and 12274392). L.W. acknowledges the support from the National Natural Science Foundation of China (Grant Nos. 12374202 and 12004001), Anhui Projects (Grant Nos. 2022AH020009, S020218016, and Z010118169), and Hefei City (Grant No. Z020132009). We also thank the Supercomputing Center of the University of Science and Technology of China, the Hefei Advanced Computing Center, and the Beijing Super Cloud Computing Center for the computer time.

## Author contributions

N.X. designed the project. D.X. and S.Z. performed the simulations. D.X., S.Z., H.T., L.W. and N.X. analyzed the data and wrote the paper. L.W. and N.X. supervised the project.

## Competing interests

The authors declare no competing interests.
