## [Peer Review File · Nature Communications]

REVIEWER COMMENTS

Reviewer #1 (Remarks to the Author):

The manuscript “Low-frequency density of vibrational states of ordinary and ultra-stable glasses” by Ding Xu et al. performed a comprehensive numerical study and theoretical analysis to understand the low-frequency vibration properties of non-phonon modes in disordered solids, which is a fundamental scientific problem crucially important in understanding the low-temperature phenomena of disordered solids. Numerous recent theoretical analyses have proposed that the low-frequency non-phonon modes obey a power-law scaling. However, the exact value of the scaling exponent has been highly controversial. The authors have performed state-of-the-art numerical simulations to resolve this highly debated mystery. By categorizing the disordered solids into two groups of unstable and stable glasses, the authors discovered two distinct scaling behaviors associated with the low-frequency non-phonon modes, with the scaling associated with unstable glasses independent of spatial dimensions and the one related to stable glasses showing spatial-dimension dependence. A general vibration density of states of the non-phonon modes thus can be described as the superposition of the two, as quantitatively expressed in equation (1) in the manuscript. Their results thus clarified the previous confusion regarding the range of scaling exponents seen in the literature. Thus, the present work represents a significant step towards understanding the low-temperature properties of disordered solids. For this reason, I strongly recommend the manuscript be accepted for publication in Nature Communications.

The following is a list of suggestions the authors may consider to improve the manuscript further.

(1) In the section on Methods, the authors have discussed the extended Hessian matrix. However, how the extended Hessian matrix is constructed still needs to be determined. The authors have included $n_{ex}=(d+1)d/2$ degrees of freedom to incorporate boundary deformation. For instance, when the dimension $d=2$, $n_{ex}=3$, representing one isotropic compression or expansion and two shear deformations. In considering the change of the energy U concerning these three deformation modes, do the authors consider only the uniform affine deformation? Specifically, in the case of the mode of isotropic compression or expansion, are the particle displacements directly proportional to the individual particle's position vector concerning the center of the system? Such an affine deformation will cause mechanical nonequilibrium for disordered solids for the individual particles. Hence, the nonaffine displacement field is usually included. It might be helpful if the authors could include more details.

(2) In analyzing the vibration properties of the non-phonon modes, separating the contributions of phonon-like and non-phonon modes is necessary. It will be helpful if the authors can include the details on how these two modes are separated in the Methods section.

(3) The authors have shown that for unstable glasses $\alpha_u=3.3$, which is systematically below the exponent $\alpha=3$ predicted by certain theoretical models and $\alpha=d+1$ for $d=2$ dimensions predicted by the fluctuating modulus theory. It might be beneficial to have a discussion here regarding this point.

(4) When there are two scaling exponents, as shown in equation (1) in the manuscript, it is intuitive that the term proportional to ω^{α_s} wins in sufficiently large systems since $\alpha_s > \alpha_u$, consistent with the Fig. 2. Moreover, the finite size analysis suggests that

ω_d and ω_s vanish in the thermodynamic limit. Therefore, only the scaling ω^4 may be left. It will be very helpful if the authors can include a discussion of the experimental measurements of molecular glasses using neutron or X-ray or other techniques where the systems are very close to the thermodynamic limit.

(5) It might be helpful if the authors could include some visual comparison of the spatial patterns of the modes between the unstable and stable glasses so that readers can gain some physical intuition.

(6) Including the fitting error bars for the scaling exponents of α_s and α_u will be useful.

Reviewer #2 (Remarks to the Author):

The authors examined the scaling of the density of harmonic vibrational modes below the first phonon band in glass forming liquids. This is a technique is used to determine the scaling of non-phononic modes in simulated glasses. While conceptually straight forward, there are many practical problems that have emerged in using this technique. It has been argued that it suffers from finite size effects, it takes a large number of glass configurations to obtain good statistics, and one rarely observes power law scaling over decades of frequency and density. The authors divide their glass configurations into stable and unstable glasses using an extended Hessian where they include deformation. They argue that the scaling of the density of harmonic vibrational modes of the unstable glasses goes as $\omega^{3.3}$ and that the stable modes scales as $\omega^{5.5}$ in 2D and $\omega^{6.5}$ in 3D. Importantly, they show that there is a system size dependence fraction of the configurations that are stable, and this would naturally explain several previous results presented in the literature. While these observations are important, there are several issues with the manuscript and I believe the authors need to do more work to make the research complete.

The author need to characterize the modes in the low-frequency scaling regime for the unstable and the stable glasses. It is generally found that the low-frequency modes are quasi-localized vibrational modes that are weakly hybridized with plane waves. What is the participation ratio of the low-frequency modes and how do these change for the stable and unstable glasses? For the density of unstable modes shown in Figure 1 there is a distinct peak at low frequencies that appears to be below the first phonon band. What is the origin of this peak?

The authors also need to indicate the unstable direction given by the negative eigenvalue of the Hessian. I believe this unstable direction can only be related to shear deformations, but it would be interesting if I was wrong.

When one argues for a scaling collapse of the density of states, one must rescale the x-axis and y-axis, otherwise the result is no longer a density and the integral of the resulting function is not one. In Figure 3 it appears that only the x-axis is rescaled, but not the y-axis. The y-axis needs to be rescaled by the determinant of the Jacobian. This makes Figure 4 confusing since the rescaling of the y axis should be $N^{-0.65} = N^{1.54}$ and not $N^{2.6}$. They make the argument that $N^{2.6}$ is correct due to a scaling argument that holds for functions, but they use this argument for densities and the argument is not valid. These scalings need to be re-examined and most likely dropped.

The authors state that "the soft potential model theory predicts $\alpha = 3$ conditionally." and give Ref. [52] as evidence. This is not a correct statement. The soft potential model ("theory" should be dropped) predicts $\alpha = 4$, and the arguments in Ref. [52] only loosely pertain to the soft potential model. This statement and the resulting discussion on $\alpha = 3$ pertaining to the soft potential model should be dropped. For a clear discussion on the soft potential model, I recommend the book "Low-Temperature Thermal and Vibrational Properties of Disordered Solids" edited by Ramos, and the original source is Phys. Stat. Sol.(a) 135, 477 (1993).

The existence of a region of ω^4 scaling in Figure 4 is not convincing. The scaling appears to be approximately valid for less than a decade in ω and only for one system size. What happens when the authors examine larger system sizes? To me it appears that the ω^4 scaling is an artifact of crossing over from $\omega^{6.5}$ scaling to ω^2 scaling. This makes the conclusion that the ω^4 scaling survives in 3D systems in the thermodynamic limit questionable.

In the introduction the authors state that the non-phonon modes have been successfully applied to understand energy transport. This is a little misleading. The modes that the authors are talking about that pertain to energy transport in the given references would be above the Ioffe-Regel frequency, and the modes studied in the submitted manuscript are below the Ioffe-Regel frequency.

The acronym DOVS is odd. Normally it is just DOS (density of states) or VDOS (vibrational density of states).

The temperature scaling of the thermal conductivity is believed to be governed by the phonon mean free path and the specific heat at low temperatures. For temperatures where the mean free path is equal to the size of crystalline domains for a crystal, then one gets the same scaling for the thermal conductivity as the specific heat since the mean free path is constant. The authors have brushed aside this detail in the introduction. The anomalous thermal conductivity below 1K of glasses is believed to be due to the change of the phonon mean free path due to two-level systems, and not due to the scaling of the low-frequency density of states. Additionally, the linear in temperature dependence of the specific heat at low frequencies is not due to the density of states studied in this manuscript, but rather the constant density of states of two-level systems. For these reasons, I find that the introduction is not relevant for the manuscript, and should be rewritten.

The authors have made a very interesting observation about a procedure frequently used to obtain the scaling of non-phononic modes in amorphous solids. Unfortunately, there is incorrect and incomplete analysis, as well as incorrect description of models and the physics. I encourage the authors to carefully modify the manuscript to remove these errors and resubmit.

Response to Reviewers' Comments

Response to Reviewer #1's Comments

1. *“The manuscript “Low-frequency density of vibrational states of ordinary and ultra-stable glasses” by Ding Xu et al. performed a comprehensive numerical study and theoretical analysis to understand the low-frequency vibration properties of non-phonon modes in disordered solids, which is a fundamental scientific problem crucially important in understanding the low-temperature phenomena of disordered solids. Numerous recent theoretical analyses have proposed that the low-frequency non-phonon modes obey a power-law scaling. However, the exact value of the scaling exponent has been highly controversial. The authors have performed state-of-the-art numerical simulations to resolve this highly debated mystery. By categorizing the disordered solids into two groups of unstable and stable glasses, the authors discovered two distinct scaling behaviors associated with the low-frequency non-phonon modes, with the scaling associated with unstable glasses independent of spatial dimensions and the one related to stable glasses showing spatial-dimension dependence. A general vibration density of states of the non-phonon modes thus can be described as the superposition of the two, as quantitatively expressed in equation (1) in the manuscript. Their results thus clarified the previous confusion regarding the range of scaling exponents seen in the literature. Thus, the present work represents a significant step towards understanding the low-temperature properties of disordered solids. For this reason, I strongly recommend the manuscript be accepted for publication in Nature Communications.”*

Response:

We are grateful to the reviewer for the precise summary of our major findings and recommending publication of our manuscript. We would also like to thank the reviewer for the suggestions and comments to help improve our manuscript. We have carefully addressed all the reviewer's concerns and made corresponding changes to the manuscript. Below is our point-by-point response to the reviewer's comments. We sincerely hope that the reviewer could find our response and revision acceptable.

2. *“following is a list of suggestions the authors may consider to improve the manuscript further. (1) In the section on Methods, the authors have discussed the extended Hessian matrix. However, how the extended Hessian matrix is constructed still needs to be determined. The authors have included $n_{ex}=(d+1)d/2$ degrees of freedom to incorporate boundary deformation. For instance, when the dimension $d=2$, $n_{ex}=3$, representing one isotropic compression or expansion and two shear deformations. In considering the change of the energy U concerning these three deformation modes, do the authors consider only the uniform affine deformation? Specifically, in the case of the mode of isotropic compression or expansion, are the particle displacements directly proportional to the individual particle's position vector concerning the center of the system? Such an affine deformation will cause mechanical nonequilibrium for*

disordered solids for the individual particles. Hence, the nonaffine displacement field is usually included. It might be helpful if the authors could include more details.”

Response:

The reviewer is right that the inclusion of the n_{ex} degrees of freedom involves compression and shear. These degrees of freedom are actually compression and shear strains from the upper triangular elements of the strain tensor. In the revised manuscript, we have added more details about the definitions of these degrees of freedom (lines 469-475 and Eq. 6). We hope that it could help the readers better understand the extended Hessian matrix. The extended Hessian matrix is exactly the same as used in Ref. 58 (Phys. Rev. E 90, 022138 (2014)). We thus refer to this reference in the revised manuscript, which contains more detailed discussions of the extended Hessian matrix and its role in the stability analysis. The reviewer is also right that when calculating the derivative of the energy with respect to the compression or shear the affine deformation is naturally imposed. However, the nonaffine displacement field is just the result of the Hessian matrix if compression or shear is in fact applied, in order to maintain force balance. It cannot be known in advance and involved in the calculation of the Hessian matrix.

3. “2) *In analyzing the vibration properties of the non-phonon modes, separating the contributions of phonon-like and non-phonon modes is necessary. It will be helpful if the authors can include the details on how these two modes are separated in the Methods section.”*

Response:

This is indeed a realistic problem when phonon-like and non-phonon modes mix up. There have been mainly three different approaches in the literature to deal with this problem: (i) separating modes by their participation ratio under the assumption that the low-frequency non-phononic modes possess obviously smaller value than phonon-like modes, e.g., PNAS 114, E9767 (2017) and Nat. Commun. 10, 26 (2019), (ii) studying non-phononic modes of small systems below the lowest Goldstone mode (i.e., lowest-frequency phonon-like mode), e.g., Phys. Rev. Lett. 117, 035501 (2016) and Phys. Rev. Lett. 121, 055501 (2018), and (iii) suppressing the influence of phonon-like modes through particle pinning, e.g., PNAS 115, 8700 (2018) and Phys. Rev. E 106, 054611 (2022). In our study, we employ approach (ii). As shown in the modified Fig. 1 of the revised manuscript, we discuss the scaling behaviors below the first Goldstone mode, so there is no interference from the phonon-like modes yet.

4. “3) *The authors have shown that for unstable glasses $\alpha_u=3.3$, which is systematically below the exponent $\alpha=3$ predicted by certain theoretical models and $\alpha=d+1$ for $d=2$ dimensions predicted by the fluctuating modulus theory. It might be beneficial to have a discussion here regarding this point.”*

Response:

Because $\alpha_u = 3.3$ is independent of spatial dimension, it should not agree with theories predicting d -dependent exponents, such as the fluctuating elasticity theory. We connect α_u to the

prediction from fold instability because by definition unstable solids are close to instabilities. The exponent 3 predicted by fold instability (Ref. 36, Phys. Rev. Lett. 119, 215502 (2017)) is based on the assumption that the distribution of the stress distance to the instability, $P(\tau)$, is constant. However, this assumption is rough. Our ongoing work with Horst-Holger Boltz, Andrea J. Liu, and Sidney R. Nagel indicates that a more careful fitting to the distribution results in a Weibull distribution, $P(\tau) \sim \tau^\theta e^{-\tau^{1+\theta}}$, with $0 < \theta < 0.1$. Using this Weibull distribution, we can derive $\alpha = 3 + 4\theta$. Therefore, our $\alpha_u = 3.3$ could be a reasonable fold-instability value based on this update. In the revised manuscript, we have added a short discussion about this (lines 182-191). Moreover, in the new Fig. 2 of the revised manuscript, we compare the lowest-frequency mode of an unstable solid with its unstable mode of the extended Hessian matrix. They are almost identical. From the fold instability argument, the structure of the modes contributing to the ω^3 scaling remains unchanged on approaching the instability. We think that the agreement between the two modes shown in Fig. 2 is another evidence of our fold instability argument about α_u (lines 222-245).

5. *“4) When there are two scaling exponents, as shown in equation (1) in the manuscript, it is intuitive that the term proportional to ω^{α_s} wins in sufficiently large systems since $\alpha_s > \alpha_u$, consistent with the Fig. 2. Moreover, the finite size analysis suggests that ω_d and ω_s vanish in the thermodynamic limit. Therefore, only the scaling ω^4 may be left. It will be very helpful if the authors can include a discussion of the experimental measurements of molecular glasses using neutron or X-ray or other techniques where the systems are very close to the thermodynamic limit.”*

Response:

We also expect that stable solids dominate in sufficiently large systems, at least because the fraction of stable solids approaches almost one. After considering Reviewer #2's comments, we admit that our current results may not be convincing enough to conclude that only the ω^4 scaling may be left in the thermodynamic limit. Instead, we would like to conservatively propose it to be a possibility and have made corresponding changes to the manuscript. Experimental measurements using neutron or X-ray have greatly improved our understanding of molecular glasses, especially the characteristics of the constituent modes of the boson peak. However, in very large systems, phonon-like modes would inevitably mix up with non-phononic modes. It remains unclear whether experimental measurements can successfully separate these two kinds of modes, especially at energy levels much lower than the boson peak. As far as we know, there have been no convincing reports of the scaling exponent below the boson peak in experiments of molecular glasses. In the revised manuscript, we have added some discussions of the experimental measurements (lines 51-54 and lines 93-96).

6. *“5) It might be helpful if the authors could include some visual comparison of the spatial patterns of the modes between the unstable and stable glasses so that readers can gain some physical intuition.”*

Response:

In the revised manuscript, we have added a new figure (Fig. 2) to compare the mode structures of 2D stable and unstable solids in the scaling regimes. Discussions about Fig. 2 are from line 222 to 245.

7. *“6) Including the fitting error bars for the scaling exponents of α_s and α_u will be useful.”*

Response:

We estimated the fitting errors and added them to lines 180 and 181.

Response to Reviewer #2's Comments

1. *“The authors examined the scaling of the density of harmonic vibrational modes below the first phonon band in glass forming liquids. This is a technique is used to determine the scaling of non-phononic modes in simulated glasses. While conceptually straight forward, there are many practical problems that have emerged in using this technique. It has been argued that it suffers from finite size effects, it takes a large number of glass configurations to obtain good statistics, and one rarely observes power law scaling over decades of frequency and density. The authors divide their glass configurations into stable and unstable glasses using an extended Hessian where they include deformation. They argue that the scaling of the density of harmonic vibrational modes of the unstable glasses goes as $\omega^{3.3}$ and that the stable modes scales as $\omega^{5.5}$ in 2D and $\omega^{6.5}$ in 3D. Importantly, they show that there is a system size dependence fraction of the configurations that are stable, and this would naturally explain several previous results presented in the literature. While these observations are important, there are several issues with the manuscript and I believe the authors need to do more work to make the research complete.”*

Response:

We are grateful to the reviewer for precisely summarizing the existing challenges of the research topic and the major findings and novelty of our work. We would also like to thank the reviewer for the criticisms and suggestions to improve our manuscript. We admit that there were problems with our data analysis and writing, as criticized by the reviewer. Following the reviewer's criticisms and suggestions, we have made significant changes to the manuscript. Below is our point-by-point response to the reviewer's comments. We sincerely hope that the reviewer could find our response and revision acceptable.

2. *“The author need to characterize the modes in the low-frequency scaling regime for the unstable and the stable glasses. It is generally found that the low-frequency modes are quasi-localized vibrational modes that are weakly hybridized with plane waves. What is the participation ratio of the low-frequency modes and how do these change for the stable and unstable glasses? For the density of unstable modes shown in Figure 1 there is a distinct peak at low frequencies that appears to be below the first phonon band. What is the origin of this peak?”*

Response:

In the revised manuscript, we have added new panels (c and d) to Fig. 1 to show the participation ratio and a new figure (Fig. 2) to show 2D examples of the mode structure for stable and unstable solids. The modes in the low-frequency scaling regime are still quasi-localized. In average, the participation ratio decreases with the decrease of frequency. Seen from the participation ratio and mode structure, there are no apparent differences between stable and unstable solids, only that the frequency is lower for unstable solids. The descriptions and discussions about these new plots are in lines 213-245.

We are also puzzling about the origin of the first peak in $D_u(\omega)$ below the first phonon band. In Fig. 3d and h of the revised manuscript, we have added more curves to show the system size

dependence of the characteristic frequencies, ω_p , ω_d , and ω_s . As defined in our manuscript, ω_p , ω_d , and ω_s denote the frequency of the first peak in $D_u(\omega)$ below the first Goldstone mode, the frequency below which $D_s(\omega)$ and $D_u(\omega)$ depart from each other, and the frequency of 3D stable solids above which the ω^{α_s} scaling is deviated, respectively. As discussed in lines 270-289 of the revised manuscript, ω_d is approximately scaled with $N^{-1/d}$, similar to the Goldstone modes. We thus suspect that ω_d may be associated with some inherent properties of the solids, e.g., the elastic moduli which contribute to the Goldstone modes. However, ω_p concerned by the reviewer shows a spatial dimension independent scaling, $\omega_p(N) \sim N^{-0.55}$. In 3D, it almost collapses with ω_s

3. *“The authors also need to indicate the unstable direction given by the negative eigenvalue of the Hessian. I believe this unstable direction can only be related to shear deformations, but it would be interesting if I was wrong.”*

Response:

The reviewer is right that the unstable direction is related to shear deformations, but it is usually accompanied with the compression. In the new Fig. 2 of the revised manuscript, we illustrate a 2D example of the unstable direction of the extended Hessian matrix together with the particle polarization vector field of the mode. For this case, the ratio of the three strain values is $\epsilon_{xx}:\epsilon_{yy}:\epsilon_{xy} = -0.359:0.352:-1$, indicating the joint deformation by shear and compression. We also notice that the particle polarization vector field of the unstable mode is identical to that of the lowest mode of the normal Hessian matrix. We think this provides another evidence supporting our argument that the scaling exponent α_u of unstable solids originates from fold instability, consistent with the finding of Ref. 36 (Phys. Rev. Lett. 119, 215502 (2017)). The discussions about Fig. 2 are in lines 222-245.

4. *“When one argues for a scaling collapse of the density of states, one must rescale the x-axis and y-axis, otherwise the result is no longer a density and the integral of the resulting function is not one. In Figure 3 it appears that only the x-axis is rescaled, but not the y-axis. The y-axis needs to be rescaled by the determinant of the Jacobian. This makes Figure 4 confusing since the rescaling of the y axis should be $N^{-0.65} = N^{1.54}$ and not $N^{2.6}$. They make the argument that $N^{2.6}$ is correct due to a scaling argument that holds for functions, but they use this argument for densities and the argument is not valid. These scalings need to be re-examined and most likely dropped.”*

Response:

We agree with the reviewer that what we did cannot conserve the integral. In the old Figs. 3 and 4, we just intended to show the system size dependence of the prefactors of the two scalings and the crossover from the $\omega^{6.5}$ to ω^4 scaling for 3D systems, respectively. We accept the reviewer’s criticism and have replotted the old Fig. 3 (now Fig. 4 of the revised manuscript) by rescaling the y-axis as well and dropped the scaling of the old Fig. 4 (now Fig. 5 of the revised manuscript). The corresponding discussions have also been rewritten (lines 290-297).

5. *“The authors state that “the soft potential model theory predicts $\alpha = 3$ conditionally.” and give Ref. [52] as evidence. This is not a correct statement. The soft potential model (“theory” should be dropped) predicts $\alpha = 4$, and the arguments in Ref. [52] only loosely pertain to the soft potential model. This statement and the resulting discussion on $\alpha = 3$ pertaining to the soft potential model should be dropped. For a clear discussion on the soft potential model. I recommend the book “Low-Temperature Thermal and Vibrational Properties of Disordered Solids” edited by Ramos, and the original source is Phys. Stat. Sol.(a) 135, 477 (1993).”*

Response:

We have carefully studied the book and the paper suggested by the reviewer and agree that we made the incorrect statement about the soft potential model. Thank the reviewer for pointing out our incorrect statement. We have dropped the statement from the manuscript.

6. *“The existence of a region of ω^4 scaling in Figure 4 is not convincing. The scaling appears to approximately valid for less than a decade in ω and only for one system size. What happens when the authors examine larger system sizes? To me it appears that the ω^4 scaling is an artifact of crossing over from $\omega^{6.5}$ scaling to ω^2 scaling. This makes the conclusion that the ω^4 scaling survives in 3D systems in the thermodynamic limit questionable.”*

Response:

We agree with the reviewer that the ω^4 scaling is not convincing based on current results. In Fig. 5a of the revised manuscript (replotted Fig. 4 of the old manuscript with the finite-size scaling being dropped), we add the $D_s(\omega)$ curve of a larger system size ($N = 6000$) and see that the range of the ω^4 scaling still shows the trend to extend slightly. Since it seems to become more pronounced when system size increases, especially for ultra-stable glasses it already emerges above ω_s when the system size is still small (see Fig. 5b of the revised manuscript), it might not be plausible to simply ignore and exclude it. Therefore, we still wish to leave the ω^4 scaling as a possibility. In the revised manuscript, we have weakened all statements saying it is (or ought to be) ω^4 , leaving the question about whether it is indeed ω^4 or just a crossover to be attacked by future studies of much larger systems, which definitely requires a sufficiently large computational capacity. Moreover, we stress in the revised manuscript that the ω^4 scaling does not emerge in $D_s(\omega)$ in 2D, indicating that the ω^4 scaling does not generally exist, unlike being argued by early studies.

We also notice that the higher frequency part right above the unconvincing ω^4 regime is not exactly ω^2 . The fitting of our largest system size results gives approximately an $\omega^{1.5}$ scaling. It is also a question whether it may evolve to ω^2 in the large system size limit. In Fig. 5 of the revised manuscript, we have also added the $\omega^{1.5}$ scaling.

In the revised manuscript, we have made significant changes to the discussions about the scalings shown in Fig. 5 (see lines 298-325).

7. *“In the introduction the authors state that the non-phonon modes have been successfully applied*

to understand energy transport. This is a little misleading. The modes that the authors are talking about that pertain to energy transport in the given references would be above the Ioffe-Regel frequency, and the modes studied in the submitted manuscript are below the Ioffe-Regel frequency.”

Response:

The reviewer is right that the references mainly discuss the energy transport of the excess modes beyond the Ioffe-Regel frequency. Thank the reviewer for this correction. We have removed the statement to avoid misleading the readers.

8. *“The acronym DOVS is odd. Normally it is just DOS (density of states) or VDOS (vibrational density of states).”*

Response:

We have changed DOVS to VDOS.

9. *“The temperature scaling of the thermal conductivity is believed to be governed by the phonon mean free path and the specific heat at low temperatures. For temperatures where the mean free path is equal to the size of crystalline domains for a crystal, then one gets the same scaling for the thermal conductivity as the specific heat since the mean free path is constant. The authors have brushed aside this detail in the introduction. The anomalous thermal conductivity below 1K of glasses is believed to be due to the change of the phonon mean free path due to two-level systems, and not due to the scaling of the low-frequency density of states. Additionally, the linear in temperature dependence of the specific heat at low frequencies is not due to the density of states studied in this manuscript, but rather the constant density of states of two-level systems. For these reasons, I find that the introduction is not relevant for the manuscript, and should be rewritten.”*

Response:

We agree with the reviewer that we did not make precise statements about the relation between the vibrational density of states and thermal properties of low temperature glasses, especially in temperatures below 1K where two-level systems dominate. We have rewritten the first two paragraphs of the introduction to make all points commented by the reviewer clearer and more relevant.

10. *“The authors have made a very interesting observation about a procedure frequently used to obtain the scaling of non-phononic modes in amorphous solids. Unfortunately, there is incorrect and incomplete analysis, as well as incorrect description of models and the physics. I encourage the authors to carefully modify the manuscript to remove these errors and resubmit.”*

Response:

We are grateful to the reviewer for acknowledging that our work is interesting and for all criticisms and suggestions. We have tried hard to revise the manuscript following the reviewer's comments. We sincerely hope that the reviewer could find our manuscript improved and recommend publication.

REVIEWERS' COMMENTS

Reviewer #1 (Remarks to the Author):

I have thoroughly reviewed all the materials prepared by Ding Xu and their colleagues in their revised manuscript. The authors have diligently addressed the concerns raised by both referees to my satisfaction, making substantial revisions that significantly enhance the quality and clarity of their work. The manuscript has evolved into an exemplary piece, and I am delighted to recommend its publication in Nature Communications.

One additional suggestion pertains to the new Figure 2, where improvements can be made to enhance the visibility of the arrows. In the current version, it is challenging to discern the arrows in the three panels (a-c) clearly when printed on A4 paper, unless the figure is considerably magnified on a computer screen.

Reviewer #2 (Remarks to the Author):

I appreciate the authors taking the time and effort to address my concerns. I find that they addressed all my concerns very convincingly and they addressed the concerns of the other reviewer. This work is going to change how researchers approach the calculation of normal modes and how we think about these modes. I recommend publication.

I only have one question to pose to the authors, which does not need to be answered for publication. That is, do the glasses the authors classify as unstable exist in experiments? It seems to me that they only exist on the computer due to the periodic boundary conditions and they would immediately shatter in the real world. Just curious about the authors thoughts on this question.

Response to Reviewers' Comments

Response to Reviewer #1's Comments

1. *"I have thoroughly reviewed all the materials prepared by Ding Xu and their colleagues in their revised manuscript. The authors have diligently addressed the concerns raised by both referees to my satisfaction, making substantial revisions that significantly enhance the quality and clarity of their work. The manuscript has evolved into an exemplary piece, and I am delighted to recommend its publication in Nature Communications."*

Response:

We are grateful to the reviewer for recommending publication of our manuscript.

2. *"One additional suggestion pertains to the new Figure 2, where improvements can be made to enhance the visibility of the arrows. In the current version, it is challenging to discern the arrows in the three panels (a-c) clearly when printed on A4 paper, unless the figure is considerably magnified on a computer screen."*

Response:

We have optimized the visibility of the vectors in Fig. 2.

Response to Reviewer #2's Comments

1. *"I appreciate the authors taking the time and effort to address my concerns. I find that they addressed all my concerns very convincingly and they addressed the concerns of the other reviewer. This work is going to change how researchers approach the calculation of normal modes and how we think about these modes. I recommend publication."*

Response:

We are grateful to the reviewer for recommending publication of our manuscript.

2. *"I only have one question to pose to the authors, which does not need to be answered for publication. That is, do the glasses the authors classify as unstable exist in experiments? It seems to me that they only exist on the computer due to the periodic boundary conditions and they would immediately shatter in the real world. Just curious about the authors thoughts on this question."*

Response:

In our view, there should be no unstable glasses in real systems with free boundaries. However, from our definition, we expect that unstable glasses could exist in confined experimental systems. As discussed in our manuscript (lines 397-400), we notice that an experimental study [Ref. 68, Nat. Commun. 13, 3649 (2022)] reported an ω^3 scaling of the vibrational density of states in quasi-2D nanoconfined solids. We think that it may be an experimental example of unstable glasses.